# Quantitative Spatial Characterization of Lymph Node Tumor for N Stage Improvement of Nasopharyngeal Carcinoma Patients

**DOI:** 10.3390/cancers15010230

**Published:** 2022-12-30

**Authors:** Jiang Zhang, Xinzhi Teng, Saikit Lam, Jiachen Sun, Andy Lai-Yin Cheung, Sherry Chor-Yi Ng, Francis Kar-Ho Lee, Kwok-Hung Au, Celia Wai-Yi Yip, Victor Ho-Fun Lee, Zhongshi Lin, Yongyi Liang, Ruijie Yang, Ying Han, Yuanpeng Zhang, Feng-Ming (Spring) Kong, Jing Cai

**Affiliations:** 1Department of Health Technology and Informatics, The Hong Kong Polytechnic University, Hong Kong SAR, China; 2Department of Biomedical Engineering, Faculty of Engineering, The Hong Kong Polytechnic University, Hong Kong SAR, China; 3Research Institute for Smart Ageing, The Hong Kong Polytechnic University, Hong Kong SAR, China; 4Department of Clinical Oncology, Queen Mary Hospital, Hong Kong SAR, China; 5Department of Clinical Oncology, Queen Elizabeth Hospital, Hong Kong SAR, China; 6Department of Clinical Oncology, The University of Hong Kong, Hong Kong SAR, China; 7Shenzhen Institute for Drug Control (Shenzhen Testing Center of Medical Devices), Shenzhen 518057, China; 8Department of Radiation Oncology, Peking University Third Hospital, Beijing 100191, China; 9Department of Clinical Oncology, The University of Hong Kong-Shenzhen Hospital, Shenzhen 518009, China; 10Department of Medical Informatics, Nantong University, Nantong 226007, China

**Keywords:** nasopharyngeal carcinoma, N stage, lymph node tumor, tumor geometry

## Abstract

**Simple Summary:**

The N staging system for Nasopharyngeal Carcinoma (NPC) is constantly improving for better survival risk stratification with accumulating clinical evidence. Discovering new prognostic factors often depends on clinical observations, which often lack comprehensiveness and precision. This study aimed to propose new quantitative spatial characterizations of LN tumor and demonstrate their feasibility of improving N stage. Independent anatomical prognostic factors were discovered and achieved superior risk stratification performance when combined with N stage. This quantitative approach could be applied to other cancer sites to discover new prognostic or predictive factors and ultimately benefit precision medicine.

**Abstract:**

This study aims to investigate the feasibility of improving the prognosis stratification of the N staging system of Nasopharyngeal Carcinoma (NPC) from quantitative spatial characterizations of metastatic lymph node (LN) for NPC in a multi-institutional setting. A total of 194 and 284 NPC patients were included from two local hospitals as the discovery and validation cohort. Spatial relationships between LN and the surrounding organs were quantified by both distance and angle histograms, followed by principal component analysis. Independent prognostic factors were identified and combined with the N stage into a new prognostic index by univariate and multivariate Cox regressions on disease-free survival (DFS). The new three-class risk stratification based on the constructed prognostic index demonstrated superior cross-institutional performance in DFS. The hazard ratios of the high-risk to low-risk group were 9.07 (*p* < 0.001) and 4.02 (*p* < 0.001) on training and validation, respectively, compared with 5.19 (*p* < 0.001) and 1.82 (*p* = 0.171) of N3 to N1. Our spatial characterizations of lymph node tumor anatomy improved the existing N-stage in NPC prognosis. Our quantitative approach may facilitate the discovery of new anatomical characteristics to improve patient staging in other diseases.

## 1. Introduction

Nasopharyngeal carcinoma (NPC) has a high prevalence in southeast Asia [1,2]. With the development of the intensity modulated radiation therapy (IMRT) technique, better survival patterns can be achieved for patients with early and late stage NPC, especially local and regional tumor control [3,4]. However, distant metastasis remained the primary failure pattern with a high occurrence rate in five years for patients with advanced lymph node (LN) metastasis [5,6]. In addition, nodal metastasis is associated with poor prognosis in other head-and-neck cancer (HNC) subtypes, such as paranasal squamous cell carcinoma [7]. Thus, effective prognosis stratification, especially for the LN tumor, is necessary to guide more accurate clinical decision-making for personalized treatments [8,9].

N stage, which belongs to the tumor–node–metastasis (TNM) staging system jointly proposed by the American Joint Committee on Cancer (AJCC) and the Union for International Cancer Control (UICC), is one of the most robust and widely used LN classifications [10]. The current edition (8th) for NPC is based on anatomical characterization, including size, laterality, and location. However, N stage has been suggested to be less comprehensive and precise due to the qualitative definitions [11].

Over the past decades, various new LN anatomical descriptors have been proposed to improve the current N staging system [12]. For instance, parotid lymph node (PLN) involvement was found to be associated with a poor prognosis in distant metastasis, and an upgrade to the N3 classification was recommended [13,14]. Besides, the current N-staging system categorizes retropharyngeal lymph node (RLN) involvement (≤6 cm) as N1 disease. However, Huang et al. suggested an upgrade of patients with bilateral retropharyngeal lymph node involvement to N2 due to the distinctive prognostic performance within N1 [15]. Other anatomical characteristics of LN, such as extra-nodal extension [16,17,18] and positive LN numbers [11,19] have been proposed to improve the existing N stage classification system for NPC.

Despite the tremendous efforts made, the development of a more accurate N staging system was still hindered by the rather complex LN anatomical environment. In the era of IMRT, detailed tumor and normal tissue delineations have become the standard procedure for treatment planning with the increasing availability of advanced imaging techniques such as MRI and PET [20,21,22]. Quantitative spatial characterization of metastatic LN may provide more accurate descriptions of its anatomy, enabling the holistic discovery of anatomical prognostic factors by a data-driven approach.

Therefore, this study aims to investigate the feasibility of improving the prognosis stratification of N staging system from quantitative spatial characterizations of metastatic LN. We designed two types of geometric histograms based on the distances and angles of LN tumor volume to surrounding normal tissues. Independent prognostic factors were extracted by principal component analysis and combined into one prognostic index. A new risk stratification from the combined index was proposed and evaluated on multiple survival endpoints, including disease-free survival (DFS), overall survival (OS), relapse-free survival (RFS) and distant metastasis-free survival (DMFS) both internally and externally. Our methodology may promote accelerated improvement of the LN classification for NPC and can be potentially generalized to other cancer sites.

## 2. Materials and Methods

Two cohorts of biopsy-proven NPC patients receiving chemoradiotherapy were retrospectively recruited from Hong Kong Queen Mary Hospital (QMH) between 2013 and 2019 and Hong Kong Queen Elizabeth Hospital (QEH) between 2012 and 2015, respectively. Informed consents from patients were waived due to the retrospective nature of this study. The total number of included patients was 194 from QMH and 284 from QEH after excluding patients with (1) co-existing cancer or distance metastasis before treatment, (2) radiation therapy only without concurrent chemoradiotherapy, (3) patients in stage N0 who do not have visible tumor in the lymph node region and (4) incomplete clinical record and missing segmentations. Patients from the QMH cohort were used for deriving independent prognostic factors and development of prognostic index, while the QEH cohort was used solely for external validation.

Clinical factors, including age, sex, T stage, N stage, M stage, overall stage, chemotherapy strategy, and survival information were collected from patient folders. The time of OS, RFS, DMFS, and DFS is defined from the date of treatment to the earliest occurrence of death from any cause, local or regional tumor recurrence, distant metastasis, and the combination of above all, respectively. The TNM stage was administered according to the 7th edition of the AJCC protocol for the QEH cohort and switched to the 8th edition after 2017 for the QMH cohort. Treatment planning structure sets were retrieved from the Picture Archiving and Communication System (PACs) in Digital Imaging and Communications in Medicine (DICOM) format. The gross tumor volume in LN (GTVn) was contoured from contrast-enhanced CT fused with MRI in QEH and an extra imaging modality of PET/CT in QMH by oncologists with at least five years of experience.

Distance and angle histograms were designed to describe the spatial configuration of GTVn relative to the surrounding organs at risk (OARs). OARs that were consistently delineated across the two institutions, including SpinalCord, Parotids (combined Left and Right Parotid), Mandible, Larynx, and Brainstem, were included in this study. Overlap volume histogram (OVH) was first proposed by Kazhdan et al. for quantifying patient geometries [23] and successfully applied by Wu et al. to predict the optimal dose-volume histogram for knowledge-based treatment planning [24]. It summarizes the distances between OAR and the target volume by recording the fractional OAR volume as a function of the maximum distance from the PTV surface: (1)$$OVH\left(d\right)=\frac{count_{i}\left(r\left(v_{OAR}^{i},S_{GTVn}\right)\right)}{V_{OAR}},$$
where $r(v_{OAR}^{i},S_{GTVn})$ is the surface distance defined as the minimum Euclidean distance from OAR voxel $v_{OAR}^{i}$ to all the LN tumor surface points $v_{GTVn}^{k}$: (2)$$r\left(v_{OAR}^{i},S_{GTVn}\right)=\min_{k}\{∥v_{OAR}^{i}-v_{GTVn}^{k}∥|v_{GTVn}^{k}\in S_{GTVn}\}.$$

The surface distance is positive for an OAR voxel outside GTVn and negative when inside. We used the signed Euclidean distance transform algorithm [25] provided by the Python package SimpleITK (version 2.1.1) [26] to calculate the surface distance map and acquired the OVH as the cumulative histogram within the OAR mask. An example GTVn surface distance map is visualized by the heat map in Figure 1c where the left parotid (Parotid_L) is drawn as a red contour.

Spatial configuration of the lymph node tumor could not be precisely determined by distance alone due to the complex organ structures in the head-and-neck region. We designed the projection overlap volume (POV) histogram to quantify the angular relationships between GTVn and the surrounding OARs. POV is defined as the relative OAR volume that overlaps with the parallel projection of GTVn: (3)$$POV\left(\alpha \right)=\frac{\sum_{i}\chi_{\alpha i}}{V},\;\chi_{\alpha i}=f\left(x\right)=\left\{\begin{array}{cc} 1, & \mathrm{if}\min_{j}\theta_{ij}<\alpha <\max_{j}\theta_{ij}\\ 0, & \mathrm{otherwise} \end{array}\right.,$$
where *V* is the voxel volume of the OAR, and $\theta_{ij}$ is the angle from GTVn surface point $v_{j}$ to OAR voxel point $v_{i}$ on the axial plane. POV histogram is calculated by summing up the masked OAR sinogram along the angle direction. The masked OAR sinogram is the modified radon transform of the OAR mask volume around the axial axis; only the voxels located before GTVn are counted for each OAR mask volume projection. One Parotid_L masked sinogram is shown in Figure 1d.

Dimensions of the OVH and POV histograms were further reduced by principal component analysis (PCA), where the components that explained the greatest variance across patients were highlighted. This study included the smallest number of principal components (PCs) of OVH and POV that explained 75% of the cumulative variance for each OAR. The coefficients of the principal components (PCs) were extracted as the potential prognostic factors.

Independent prognostic factors were identified from the selected PCs by univariate Cox regression on DFS followed by the covariate independency test with N stage through multivariate Cox regression. The final prognostic index was built by combining the independent prognostic factors with N stage through multivariate Cox regression and evaluated by concordance index (C-index). The confidence interval and *p*-values for baseline N stage comparison were determined by 1000-iteration bootstrapping. Risk stratification performance was assessed by Kaplan–Meier (KM) analysis, where patients were equally stratified into high (G1), median (G2), and low (G3) risk groups based on the prognostic index in the discovery cohort. The stratification thresholds were applied to the testing cohort as well for the three-grade stratification. Hazard ratios (HRs) with 95% confidence interval (95CI) and the log-rank *p*-values between risk groups were acquired from univariate Cox regression. All Cox regressions and KM analysis were implemented by the Python package lifelines (version 0.27.0) [27], and the *p*-value of 0.05 was considered significant.

## 3. Results

### 3.1. Baseline Patient Characteristics

Distributions of the baseline patient characteristics for the two cohorts were listed in Table 1. Consistent distributions of age, sex, overall stage, chemotherapy strategy, and World Health Organization (WHO) histology were found between the discovery and validation cohort. The T stage and N stage were significantly different (*p* ≤ 0.05) between the two institutions. The median follow-up time of the discovery cohort is 2.5 years and 4.6 years for the validation cohort. Of the 194 discovery patients within the follow-up period, 22 developed local recurrence, 17 with regional recurrence, 29 with distant metastases, and 25 died. The three-year DFS, OS, RFS, and DMFS rates were 72.1%, 90.0%, 82.4%, and 82.4%, respectively. In the validation cohort, 34, 25, 44, and 40 patients of 284 developed local recurrence, regional recurrence, distant metastasis, and death, and the five-year DFS, OS, RFS, DMFS are 74.3%, 94.0%, 85.0%, and 86.2%.

### 3.2. Prognostic LN Spatial Factors

Thirty-one PCs were extracted from the OVH and POV histograms in total, including four OVH PC and three POV PC of SpinalCord, five OVH PC and three POV PC of Parotids, two OVH PC and two POV PC of Brainstem, three OVH OC and three POV PC of Larynx, and four OVH PC and two POV OC of Mandible. After univariate and multivariate Cox regressions, two spatial factors including the first PC of spinal cord OVH (OVH${}_{SC,PC1}$) and the third PC of spinal cord POV (POV${}_{SC,PC3}$) were selected as independently prognostic to DFS. Between the two spatial factors, OVH${}_{SC,PC1}$ demonstrated a higher discriminability to DFS with C-index of 0.66 at discovery and 0.56 at external validation, while 0.57 at discovery and 0.54 at external validation for POV${}_{SC,PC3}$.

As listed in Table 2, POV${}_{SC,PC3}$ contributed the highest positive hazard (HR = 3.35, 95CI: 1.41–7.99), followed by the N stage (HR = 2.26, 95CI: 1.46–3.49). On the other hand, OVH${}_{SC,PC1}$ had the negative impact of survival hazard (HR = 0.63, 95CI: 0.48–0.83). Figure 2a presents the distributions of the two spatial factors of the 3-year disease and non-disease progressed patients at both discovery and validation. Patients who developed disease progression within three years had significantly lower OVH${}_{SC,PC1}$ (mean: −0.80 vs. −0.07, *p* = 0.007) and higher POV${}_{SC,PC3}$ (mean: 0.082 vs. 0.057, *p* = 0.012) at discovery, but smaller differences were found on the validation cohort (OVH${}_{SC,PC1}$: 0.46 vs. 0.74, *p* = 0.032; POV${}_{SC,PC3}$: −0.18 vs. −0.25, *p* = 0.089). Moreover, the spinal cord OVH appeared to be overall larger in the validation but smaller for the POV. After binarizing the two spatial factors by the median values in the discovery cohort, more patients in the validation cohort fell into the low-risk groups, as indicated by Figure 2b. The odds ratios were 0.30 (*p* = 0.006) for OVH${}_{SC,PC1}$ and 2.21 (*p* = 0.052) for POV${}_{SC,PC3}$ in the discovery cohort. They were less significant for OVH${}_{SC,PC1}$ (odds ratio = 0.60, *p* = 0.275) but more significant for POV${}_{SC,PC3}$ (odds ratio = 2.83, *p* = 0.004) in the validation cohort.

### 3.3. Combined Prognostic Index

The combined prognostic index had better discriminability than N stage on all the survival endpoints but showed statistical significance mainly in DFS and RFS, as reported in Table 3. C-index in DFS increased from 0.654 (training) and 0.568 (external validation) to 0.722 (training) and 0.603 (external validation) when combining the two new spatial factors with N stage. Such improvement was significant in training (*p*-value = 0.020) while much less in external validation (0.086). On the other hand, the training and validation improvements were both significant in RFS with C-index reaching 0.723 (*p*-value = 0.020) and 0.603 (*p*-value = 0.019), respectively.

Better risk stratifications were achieved by the combined prognostic index in DFS and DMFS than N stage itself, as shown by the KM curves in Figure 3. Table 4 reports the hazard ratios and the corresponding *p*-values between different risk groups as well as the three-year survival rates in DFS, OS, RFS, and DMFS. On the discovery cohort, the DFS survivals of the three new risk groups were statistically different (*p* ≤ 0.05) whereas much lower statistical significance was found between the N1 and N2 groups (*p* = 0.139). Higher hazard ratios were observed between G2 (4.49) and G3 (9.07) to G1 compared to the N stage (N1 vs. N2: 1.83, N1 vs. N3: 5.19). However, the HR was less between G3 to G2 (1.913) compared to the one between N3 to N2 (2.988). A similar trend was found in DMFS where G2 (4.11) and G3 (10.41) were better separated from G1 but worse between G2 and G3 (2.26). In the validation cohort, the HRs between G2 (DFS: 1.71, *p* = 0.021; DMFS: 1.72, *p* = 0.101) and G3 (DFS: 4.02, *p* < 0.01; DMFS: 2.93, *p* = 0.014) to G1 also increased significantly compared to that between N2 (DFS: 0.772, *p* = 0.518; DMFS: 0.552, *p* = 0.271) and N3 to N1 (DFS: 1.821, *p* = 0.171; DMFS: 1.876, *p* = 0.216) in both DFS and DMFS. Similarly, a less HR was found between G2 and G3 (DFS: 2.44, *p* = 0.006; DMFS: 1.74, *p* = 0.219) than between N2 and N3 (DFS: 2.66, *p*≤ 0.001; DMFS: 3.17, *p* = 0.001).

The remaining survival endpoints showed heterogeneous patterns under the new risk stratification (Table 4). Significant HR improvements were observed in OS, but marginal in RFS for the discovery cohort. On the other hand, RFS showed significantly higher stratification performance in the validation cohort, but no improvement in OS was observed. Moreover, the validation cohort demonstrated higher 3-year survival rates on G1 and lower on G2 for RFS and DMFS, whereas marginal improvement of 3-year survival rates was found in the discovery cohort.

### 3.4. Representative Cases

To further explain the contribution of the two anatomical factors in better identifying the risk of disease progression, we selected two representative cases from the discovery cohort with the same N stage but distinct risks based on the spatial index. The high-risk patient was classified as G1 and the low-risk one as G3, both having the same N stage (N2) and chemotherapy strategy (CCRT + ACT). The high-risk patient developed distant metastases at 32.3 months, while the low-risk patient showed no signs of disease progression for at least 34.3 months. Figure 4a presents the 2D axial masks and the 3D volumes of GTVn and three OARs for the high and low-risk patient. Anatomically, both patients had metastatic retropharyngeal LN, but a significantly larger extent of the right cervical LN tumor was observed in the high-risk patient. Meanwhile, distinct patterns of the spinal OVH and POV curves were found, as drawn in Figure 4b, where the selected PC vectors were also included. The OVH curve of the high-risk patient was significantly higher than that of the low-risk patient with the largest overlap volume difference emphasized at around the global minimum (~75 mm) of the first PC vector. The POV at the first local maximum (~25 degrees) of the PC vector was much higher in the high-risk patient, exceeding the higher POV of the low-risk patient at the second local maximum (~125 degrees).

## 4. Discussion

This study demonstrated the feasibility of discovering new prognostic factors from quantitative spatial characterization of LN tumor for better LN risk stratification with high cross-site generalizability. Two histograms precisely characterized the LN tumor anatomy by distances (OVH) and angles (POV). PCA effectively reduced the high-dimensional histograms into several informative and independent anatomical factors, and two final independent prognostic factors were discovered by Cox regressions in DFS. The prognostic index that combines the independent prognostic spatial factors and the N stage achieved better new three-level risk stratifications than the N stage itself in DFS and DMFS at both discovery and external validation.

Only the spinal cord spatial factor OVH${}_{SC,PC1}$ and POV${}_{SC,PC3}$ were identified as the independent prognostic factors to DFS. OVH${}_{SC,PC1}$ highlights the overlap of the lower spinal cord with the expansion of isotropic LN tumor by approximately 75 mm (Figure 4b), indicating a smaller axial expansion of LN. The PC vector of POV${}_{SC,PC3}$ has two peaks at around 25 and 125 degrees and reaches local minimums at 0 and 180 degrees (Figure 4b). Higher projection overlaps at the peak angles indicate more volume of LN tumor in the anterior direction of the spinal cord, whereas the valley angles suggest less involvement of the LN tumor on the lateral sides. Additionally, both factors are correlated with the axial extent of the LN tumor due to the thin cylindrical structure of the spinal cord. Such correlation was also demonstrated by the two example patients in Figure 4a where the high-risk patient with lower OVH${}_{SC,PC1}$ and higher POV${}_{SC,PC3}$ had a significantly larger axial extent of cervical LN.

Previous clinical observations on the prognostic power of the anatomy of LN tumors were highly correlated with our quantitative findings. The results of our survival analysis suggest an increased risk of disease progression with lower OVH${}_{SC,PC1}$ (adjusted HR = 0.63, 95CI: 0.48–0.83; *p*≤ 0.001) and higher POV${}_{SC,PC3}$ (adjusted HR = 3.35, 95CI: 1.41–7.99), regardless of the N stage. Their independent prognostic power could be explained by the two example patients in whom the high-risk one developed early distant metastases despite their identical N stage. As discussed in the previous paragraph, a higher prognostic index value suggests a higher axial expansion and extent of the LN tumor, which supports the ongoing discussion of the high prognostic value of the quantitative LN burden. Previous clinical studies reported the number of metastatic LN regions as an independent predictor of DMFS [11,19]. For POV${}_{SC,PC3}$, a higher value may also indicate retropharyngeal LN metastasis with a larger size or bilateral involvement. Retropharyngeal LN has also been suggested to indicate worse in DFS and DMFS [28,29]. Specifically, the size of the metastatic retropharyngeal LN with a cutoff axial diameter of 6mm has been identified as a significant prognostic factor for OS and DMFS [30,31]. It was also suggested that the bilateral involvement of the retropharyngeal lymph nodes should be upgraded to N2 disease due to the worse 5-year OS and DMFS [15]. These anatomical characteristics have been partially included in the definition of the N1 classification of the 7th and 8th N staging system [32], where metastasis is limited above the caudal border of cricoid cartilage and/or retropharyngeal lymph node(s) does not exceed 6mm in greatest dimension. Our quantitative anatomical factors may provide more precise descriptions of various LN anatomy characterizations, thus independent of the existing N stage classifications.

The two final spatial factors were predictive of three-year DFS and DMFS at both discovery and validation. However, the binarization thresholds were less generalizable from discovery to validation due to the overall different magnitudes of the spatial factor values. As a result, much higher low-risk patients were classified in the validation cohort when using the median values in the discovery as the binarization thresholds. The systematic cross-institutional variations in the spatial factor magnitudes could be attributed to the inconsistent spinal cord volume definitions, especially the starting and ending point. A higher spinal cord extent may lead to a lower relative overlap volume for both OVH and POV at the same absolute distance and angle, and the resulting PC coefficients are expected to be smaller. For clinical utility, consistent organ and tumor segmentations are important to ensure a reliable quantitative spatial characterization. Further adjustments in the spatial factor definitions for enhanced robustness are needed in future studies.

Despite the promising performance of the spatial characterization of lymph node tumors in survival prognosis, the analysis involves standardized tumor and OAR segmentations [33] as well as complex computations of distance and angle histograms for thorough characterization, which often require specific training. The potential long learning curve for clinicians may hinder the clinical application of the proposed predictors. Integration of AI-based systems for auto-segmentation [34] and dedicated calculation scripts into the existing treatment planning system could be one solution for fast implementation in daily clinical practice. On the other hand, other types of biomarkers, which are easier to implement in clinics, have been proposed as strong survival predictors for patients with NPC and other HNC diseases. Systematic inflammation indicators, which can be directly measured from blood test results, have been reported to be prognostic in multiple HNC subtypes. For example, pre-treatment neutrophil-to-lymphocyte ratio (NLR) has been investigated, and a strong statistical correlation was observed with positive neck occult metastasis in laryngeal squamous cell carcinoma [35]. Another study by Orabona et al. confirmed the independent prognostic power of the systemic immune-inflammation index (SII) and the systemic inflammation response index (SIRI) on OS of patients who received malignant salivary gland tumor surgery [36].

The constructed prognostic index results in improved risk stratifications in DFS and DMFS compared to the existing N stage both internally and externally. It is consistent with previous findings on the improved DMFS prognostication of the LN tumor region number [11] and the involvement of the retropharyngeal LN tumor [15]. Better risk stratifications on OS were only observed on the discovery cohort and RFS on the external validation cohort. Several reasons could contribute to the heterogeneous results. First, the thresholds of the prognostic index for the three-class risk classification could be suboptimal and less generalizable. The threshold optimization method for risk stratification requires a more careful design and wide validation for clinical practice. As discussed in the previous paragraph, the overall magnitudes of the spatial factors were inconsistent, which may contribute to the reduced generalizability of the prognostic index and the resulting risk groups. Second, some patient characteristics, such as stages and chemotherapy treatments, are rather different between discovery and external validation. They may affect the generalizability of the risk stratification performance due to the different baseline performances. Third, the sample sizes and follow-up durations are limited, especially in the discovery cohort. Less patients remained as uncensored samples, resulting in less reliable results. Increasing the sample size with more complete follow-up information is needed in future studies to enhance the clinical evidence of our findings.

## 5. Conclusions

This study used the distance histogram OVH and the newly proposed angle histogram POV to quantitatively characterize the anatomy of the LN tumor in relation to the surrounding spinal cord and parotids. Independent prognostic factors on DFS were discovered from the principal components of the anatomical histograms and combined with the N stage into an spatial index. It surpassed the N stage itself in risk discrimination and stratification. The proposed quantitative approach may facilitate the discovery of new anatomical characteristics in a more holistic and precise way to improve patient staging in other diseases.

## Figures and Tables

**Figure 1 cancers-15-00230-f001:**
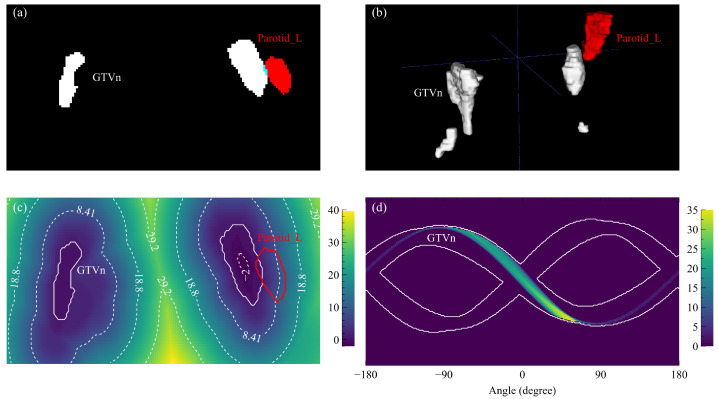
Distance and angle maps based on example GTVn and Parotid_L structures. (**a**) One axial slice of the structure masks (white: GTVn, red: Parotid_L) with the overlap region highlighted by blue. (**b**) The rendered three-dimensional structures. (**c**) One axial slice of the GTVn distance map with annotated contour lines and the Parotid_L contour. (**d**) One slice of the Parotid_L angle map masked by the GTVn sinogram edges (white contours).

**Figure 2 cancers-15-00230-f002:**
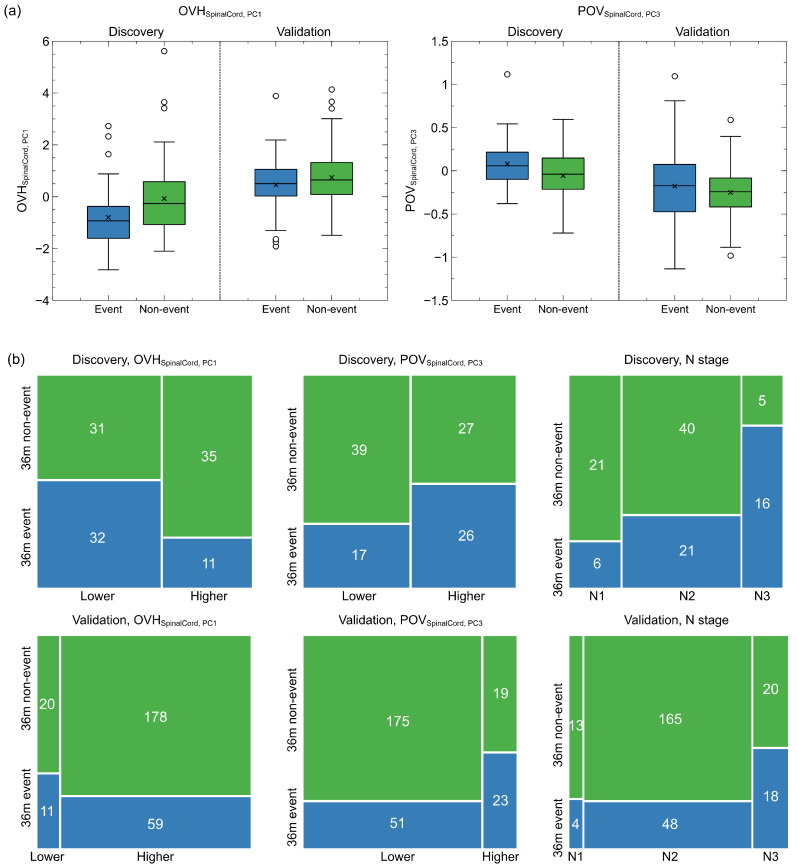
Continuous and binarized spatial factor distributions and N stage distributions for 3-year disease progressed and non-disease progressed patients in the discovery and validation cohort. (**a**) Box plots of continuous spatial factor distributions. Patients with disease progression within three years had lower mean OVH principle values and higher mean POV principle values at both discovery and validation. (**b**) Mosaic plots of the binarized spatial factor and N stage distributions of patients with and without 3-year disease progression.

**Figure 3 cancers-15-00230-f003:**
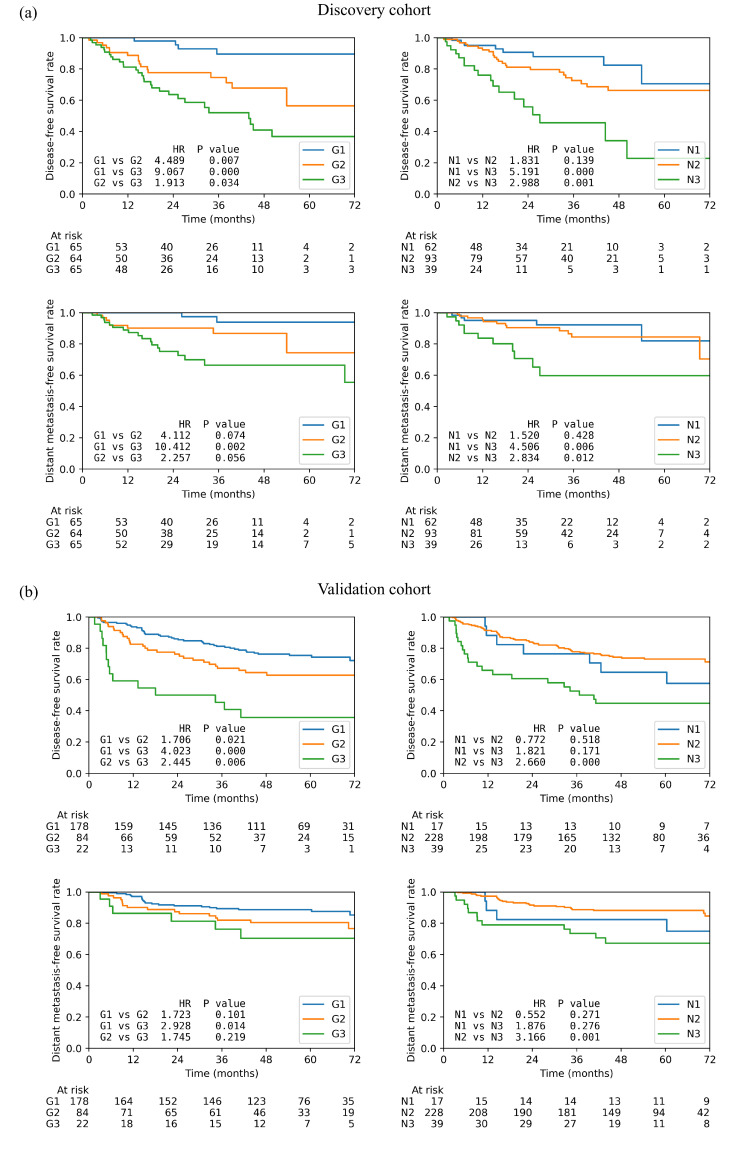
Kaplan–Meier curves of the low-(G1), median-(G2), and high-risk (G3) patient groups based on the new spatial index and the three N stages on (**a**) disease-free survival and (**b**) distant metastasis-free survival. Each plot also contains the hazard ratio (HR) and the corresponding *p*-value between each two groups.

**Figure 4 cancers-15-00230-f004:**
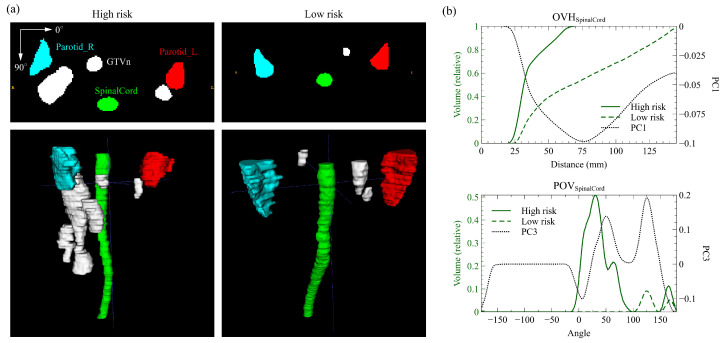
Quantitative anatomical characterizations of the high-risk and low-risk patient. (**a**) The axial slice masks and rendered 3D volumes of GTVn (lymph node tumor), Parotid_L, Parotid_R, and SpinalCord structures. (**b**) The SpinalCord overlap volume histogram (POV) and projection overlap volume (POV) of the two patients and the corresponding selected principal component (PC) vector. Significant differences in lymph node anatomy were captured by the large variations in the histograms and highlighted by the PCs.

**Table 1 cancers-15-00230-t001:** Baseline patient characteristics of the discovery and validation cohort.

		Discovery Cohort	Validation Cohort	*p*-Value
Age				
	Mean	53.39	52.16	0.249
Sex				
	Female	41	70	0.667
	Male	153	214	
N stage				
	N1	62	17	0.035
	N2	93	228	
	N3	39	39	
Chemotherapy				
	CCRT	33	178	0.330
	CCRT + ACT	78	61	
	CCRT + ICT	83	43	
WHO histology				
	Type 2	27	74	0.142
	Type 3	167	210	

Note: Staging was performed according to the 7th edition of the AJCC protocol for the validation cohort
and switched to the 8th edition after 2017 for the discovery cohort. Abbreviations: CCRT, concurrent chemoradiotherapy;
ACT, adjuvant chemotherapy; ICT, induction chemotherapy; WHO, World Health Organization.

**Table 2 cancers-15-00230-t002:** Hazard ratios and *p*-values of the selected spatial factors and N stage from multivariate Cox regression on disease-free survival.

Covariant	HR (95CI)	*p*-Value
OVH${}_{SC,PC1}$	0.63 (0.48–0.83)	<0.001
POV${}_{SC,PC3}$	3.35 (1.40–7.99)	0.006
N stage	2.26 (1.46–3.49)	<0.001

**Table 3 cancers-15-00230-t003:** Concordance index of the proposed geometric prognostic index and N stage in the training and validation cohort.

Survival Endpoint	Training Cohort			Validation Cohort		
	Prognostic Index (95CI)	N Stage (95CI)	*p*-Value	Prognostic Index (95CI)	N Stage (95CI)	*p*-Value
DFS	0.72 (0.65–0.79)	0.65 (0.57–0.73)	0.020	0.60 (0.54–0.67)	0.57 (0.52–0.62)	0.086
OS	0.75 (0.63–0.84)	0.72 (0.64–0.80)	0.245	0.60 (0.48–0.71)	0.58 (0.50–0.67)	0.395
RFS	0.72 (0.62–0.82)	0.64 (0.54–0.73)	0.020	0.60 (0.52–0.69)	0.53 (0.47–0.60)	0.019
DMFS	0.72 (0.63–0.81)	0.65 (0.54–0.76)	0.062	0.57 (0.47–0.67)	0.57 (0.50–0.65)	0.536

**Table 4 cancers-15-00230-t004:** Risk stratification performance of the proposed risk groups and N stage in multiple survival endpoints and discovery and validation cohort.

Survival Endpoint	Proposed Risk Stratification				N Stage			
	Group	HR	*p*-Value	3y SR	Group	HR	*p*-Value	3y SR
Discovery cohort								
DFS	G1	—	—	89.6%	N1	—	—	87.9%
	G2	4.49	0.007	74.6%	N2	1.83	0.139	72.6%
	G3	9.07	<0.001	52.1%	N3	5.19	<0.001	45.6%
OS	G1	—	—	97.3%	N1	—	—	100.0%
	G2	7.66	0.055	92.7%	N2	3.33	0.115	89.4%
	G3	13.98	0.011	79.7%	N3	11.62	0.002	72.6%
RFS	G1	—	—	89.6%	N1	—	—	93.0%
	G2	2.23	0.181	85.3%	N2	2.64	0.079	79.5%
	G3	4.76	0.005	72.2%	N3	4.59	0.014	74.9%
DMFS	G1	—	—	94.0%	N1	—	—	92.2%
	G2	4.11	0.074	86.8%	N2	1.52	0.428	84.5%
	G3	10.41	0.002	66.5%	N3	4.51	0.006	59.8%
Validation cohort						r		
DFS	G1	—	—	81.2%	N1	—	—	76.5%
	G2	1.71	0.021	67.2%	N2	0.77	0.518	77.8%
	G3	4.02	<0.001	45.5%	N3	1.82	0.171	52.7%
OS	G1	—	—	95.2%	N1	—	—	87.8%
	G2	1.36	0.384	93.5%	N2	1.56	0.548	95.3%
	G3	2.28	0.076	85.9%	N3	2.57	0.223	89.0%
RFS	G1	—	—	88.7%	N1	—	—	87.8%
	G2	1.46	0.219	82.9%	N2	0.84	0.736	85.9%
	G3	3.69	0.001	62.7%	N3	1.20	0.764	78.2%
DMFS	G1	—	—	89.3%	N1	—	—	82.4%
	G2	1.72	0.101	82.0%	N2	0.55	0.271	88.7%
	G3	2.93	0.014	76.2%	N3	1.88	0.276	73.5%

Note: HR and *p*-value were relative to the low-risk group (G1) or N1. Abbreviations: HR, hazard ratio; 3y SR:
3-year survival rate; DFS, disease-free survival; OS, overall survival; RFS, relapse-free survival; DMFS, distant
metastasis-free survival; 95CI: 95% confidence interval.

## Data Availability

The data presented in this study are available on request from the corresponding author. The data are not publicly available due to patient privacy protection.

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
