# Peer review of "Quantitative Spatial Characterization of Lymph Node Tumor for N Stage Improvement of Nasopharyngeal Carcinoma Patients"

_cancers, 2022, doi:10.3390/cancers15010230_

Round 1

Reviewer 1 Report

Strength of the paper: 

The search for new prognostic indicators for head and neck cancer represents a topic still discussed in the literature. This need arises from the need to improve the predictive capacity of current classification systems based on the TNM. Improving stratification would allow the creation of more targeted and effective therapeutic pathways. This is a multicenter study on a significant sample of patients collected retrospectively over a 6-year period.

Overall the work is well written and the methodology is correct. Some clarifications could improve the scientific attractiveness.

1) In the discussion section insert the current prognostic indices and the role that inflammatory biomarkers could have. NLR PLR SIRI YES.

Salzano G, Perri F, Maglitto F, Togo G, De Fazio GR, Apolito M, Calabria F, Laface C, Vaira LA, Committeri U, Balia M, Pavone E, Aversa C, Salzano FA, Abbate V, Ottaiano A, Cascella M, Santorsola M, Fusco R, Califano L, Ionna F. Pre-Treatment Neutrophil-to-Lymphocyte and Platelet-to-Lymphocyte Ratios as Predictors of Occult Cervical Metastasis in Clinically Negative Neck Supraglottic and Glottic Cancer. J Pers Med. 2021 Nov 25;11(12):1252. doi:10.3390/jpm11121252. PMID: 34945723; PMC ID: PMC8706672.

Abbate, V .; Baron, S.; Troise, S.; Laface, C.; Bonavolo, P.; Pacella, D.; Salzano, G.; Iaconetta, G.; Califano, L.; Dell'Aversana Orabona, G. The Combination of Inflammatory Biomarkers as Prognostic Indicator in Salivary Gland Malignancy. Cancers 2022,14,5934. https://doi.org/10.3390/cancers14235934

2 In the discussion section add as much as possible the complexities connected with this spatial analysis. It involves segmenting soft tissues and extracting features with a very complex procedure that requires specific training and a long learning curve.

3) Define OARs. there is only the acronym

Author Response

We thank the reviewer for the constructive comments and suggestions. A new paragraph has been added in the Discussion Section of the revised manuscript to address the first two comments:

Line 280: “Despite the promising performance of the spatial characterization of lymph node tumors in survival prognosis, the analysis involves standardized tumor and OAR segmentations [33] as well as complex computations of distance and angle histograms for thorough characterization, which often require specific training. The potential long learning curve for clinicians may hinder the clinical application of the proposed predictors. Integration of AI-based systems for auto-segmentation [34] and dedicated calculation scripts into the existing treatment planning system could be one solution for fast implementation in daily clinical practice. On the other hand, other types of biomarkers, which are easier to implement in clinics, have been proposed as strong survival predictors for patients with NPC and other HNC diseases. Systematic inflammation indicators, which can be directly measured from blood test results, have been reported to be prognostic in multiple HNC subtypes. For example, pre-treatment neutrophil-to-lymphocyte ratio (NLR) has been investigated, and a strong statistical correlation was observed with positive neck occult metastasis in laryngeal squamous cell carcinoma [35]. Another study by Orabona et al. confirmed the independent prognostic power of the systemic immune-inflammation index (SII) and the systemic inflammation response index (SIRI) on OS of patients who received malignant salivary gland tumor surgery [36]”

  1. Lee, A.W.; Ng, W.T.; Pan, J.J.; Poh, S.S.; Ahn, Y.C.; AlHussain, H.; Corry, J.; Grau, C.; Grégoire, V.; Harrington, K.J.; et al. 444 International guideline for the delineation of the clinical target volumes (CTV) for nasopharyngeal carcinoma. Radiotherapy and Oncology 2018, 126, 25–36. https://doi.org/https://doi.org/10.1016/j.radonc.2017.10.032.
  2. Lin, L.; Dou, Q.; Jin, Y.M.; Zhou, G.Q.; Tang, Y.Q.; Chen, W.L.; Su, B.A.; Liu, F.; Tao, C.J.; Jiang, N.; et al. Deep Learning for Automated Contouring of Primary Tumor Volumes by MRI for Nasopharyngeal Carcinoma. Radiology 2019, 291, 677–686, [https://doi.org/10.1148/radiol.2019182012]. PMID: 30912722, https://doi.org/10.1148/radiol.2019182012.
  3. Salzano, G.; Perri, F.; Maglitto, F.; Togo, G.; De Fazio, G.R.; Apolito, M.; Calabria, F.; Laface, C.; Vaira, L.A.; Committeri, U.; et al. Pre-Treatment Neutrophil-to-Lymphocyte and Platelet-to-Lymphocyte Ratios as Predictors of Occult Cervical Metastasis in Clinically Negative Neck Supraglottic and Glottic Cancer. Journal of Personalized Medicine 2021, 11. https://doi.org/10.3390/jpm1 1121252.
  4. Abbate, V.; Barone, S.; Troise, S.; Laface, C.; Bonavolontà, P.; Pacella, D.; Salzano, G.; Iaconetta, G.; Califano, L.; Dell’Aversana Orabona, G. The Combination of Inflammatory Biomarkers as Prognostic Indicator in Salivary Gland Malignancy. Cancers 2022, 14. https://doi.org/10.3390/cancers14235934.

We also thank the reviewer for pointing out the mistake in comment 3. We have spelled out the full name of OAR, organ at risk, for the first occurrence in the revised manuscript (line 91).

Reviewer 2 Report

Statistical comments:

1.     A rigorous approach to derive PCs should be based on the discovery cohort. That is, PC in validation cohort needs to be derived from discovery cohort. Please clarify if it is the case. If not, strongly recommend using the PC model derived from the training cohort to calculate PC in the test set.

2. Both cohorts seem not homogenous (Table 1). Will it cause bias and generate issue for model development? Will it be more appropriate to evenly distribute both cohorts to build new training and test sets. 

Author Response

Comment 1:

We thank the reviewer for pointing out the unclear information. We indeed derived the PCs from the discovery cohort only and validated the prognostic performance in both the discovery and validation cohorts, separately. The following new sentences have been added to the revised manuscript to clarify this issue.

Line 76: “Patients from the QMH cohort were used for deriving independent prognostic factors and development of prognostic index, while the QEH cohort was used solely for external validation.”

Comment 2:

We thank the reviewer for the comments and suggestions. We do realize the inhomogeneous distributions of patient characteristics between the two study cohorts. It is mainly caused by the different patient admission preferences of the two hospitals with more late-stage patients admitted to QEH. The N-stage distributions are largely correlated lymph node tumor shape, which could indeed affect the principal component analysis and thus the derived spatial risk factor. Although both cohorts seem not homogenous, it appears to be more realistic in a real-world clinical scenario. Notably, we still observed the significant improvement of survival prognostications in the external validation cohort (Table 3, 4), indicating the reliability of the proposed prognostic index even in a different patient distribution. Moreover, the derivation of the spatial risk factors could be fine-tuned for each institution for a more consistent internal performance. Therefore, we still prefer to maintain the original stage distributions of the patients in each cohort in this study in order to reflect the authentic performance of the proposed spatial factors and the final prognostic index in a real-world clinical settings. 

Reviewer 3 Report

Good work. A little introduction with the tumours of the nasal district that can do N involvment should be described. Please you can use and mention in your references this work:

Sireci F, Dispenza F, Lorusso F, Immordino A, Immordino P, Gallina S, Peretti G, Canevari FR. Tumours of Nasal Septum: A Retrospective Study of 32 Patients. Int J Environ Res Public Health. 2022 Feb 2;19(3):1713. doi: 10.3390/ijerph19031713.

Author Response

We thank the reviewer for the comment and suggestion. The following sentence has been added to the revised manuscript with the suggested reference:

Line 30: “In addition, nodal metastasis is associated with poor prognosis in other head-and-neck cancer (HNC) subtypes, such as paranasal squamous cell carcinoma [7]”

  1. Sireci, F.; Dispenza, F.; Lorusso, F.; Immordino, A.; Immordino, P.; Gallina, S.; Peretti, G.; Canevari, F. Tumours of Nasal Septum: A 368 Retrospective Study of 32 Patients. International Journal of Environmental Research and Public Health 2022, 19. Publisher Copyright: 369 © 2022 by the authors. Licensee MDPI, Basel, Switzerland., https://doi.org/10.3390/ijerph19031713.